# Blood concentrations of mediators released from activated neutrophils are related to the severity of alcohol-induced liver damage

Beata Kasztelan-Szczerbinska[1]*, Bartosz Zygo[2], Anna Rycyk-Bojarzynska[1], Agata Surdacka[3], Jacek Rolinski[3], Halina Cichoz-Lach[1]

**1** Department of Gastroenterology with Endoscopy Unit, Medical University of Lublin, Lublin, Poland,
**2** Department of Gastroenterology with Endoscopy Unit, Independent Public Academic Hospital No. 4 in Lublin, Lublin, Poland, **3** Department of Clinical Immunology, Medical University of Lublin, Lublin, Poland

* beata.szczerbinska@umlub.pl

## Abstract

### Background

Immune dysregulation and neutrophil infiltration are hallmarks of alcohol-related liver disease (ALD). Our objective was to evaluate the blood profile of neutrophil-derived mediators [neutrophil elastase (NE), myeloperoxidase (MPO), alpha1-antitrypsin (A1AT)], and their potential relevance in ALD.

### Methods

62 patients with ALD /47 males, and 15 females, aged 49,2 ± 9,9/ were prospectively recruited and distributed according to their 1/ gender, 2/ severity of liver dysfunction (by Child-Turcotte-Pugh, MELD scores, and mDF) 3/ presence of complications of ALD complications, and followed for 90 days. 24 age- and sex-matched healthy volunteers served as the control group. Neutrophil-derived biomarkers were quantified using enzyme-linked immunosorbent assays (ELISAs).

### Results

Blood concentrations of MPO and NE were significantly higher in ALD patients in comparison with controls. A1AT levels were not different. There were no gender-related differences in the studied biomarker levels. Both NE and MPO correlated with routine markers of inflammation, while NE with MELD and mDF scores. Patients with a severe ALD course i.e. MELD>20 or mDF>32, presented with significantly higher NE blood concentrations.

### Conclusions

Our results point out the critical role of neutrophils in the pathogenesis of ALD. NE and MPO correlated with the intensity of inflammation, and NE was related to the severity of liver dysfunction.

**Data Availability Statement:** All relevant data are within the paper.

**Funding:** The study was supported by research grants from the Medical University of Lublin, Poland (DS369/2018-2020). The funders had no role in the design of the study; in the collection, analyses, or interpretation of data; in the writing of the manuscript, or in the decision to publish the results.

**Competing interests:** The authors have declared that no competing interests exist.

# 1. Introduction

Neutrophils are the first-line effectors of the human innate immune system [1, 2]. Inflammatory dysregulation and neutrophil infiltration are hallmarks of alcohol-related liver disease (ALD) [3]. Given their destructive potential, extracellularly released neutrophil enzymes must be carefully controlled to prevent the detrimental effects of improperly activated neutrophils on host tissues [4–6]. A growing body of evidence has suggested that neutrophils could have an impact on ALD patient outcomes. The progression and complications of hepatic fibrosis and cirrhosis are closely related to the severity of hepatic inflammation [7]. In the majority of ALD patients, immune and inflammatory responses are up-regulated and high levels of pro-inflammatory cytokines and chemokines are present in the systemic circulation. They promote neutrophilic recruitment and liver infiltration with a subsequent release of reactive oxygen species (ROS), proteases, and inflammatory mediators that contribute to tissue injury and liver function deterioration [8, 9]. In patients with alcohol-related liver injury, prolonged or repeated activation of neutrophils may lead to their exhaustion and the development of pathogenic inflammation with immune suppression. Neutrophils lose their effective phagocytic antimicrobial skills, leading to increased susceptibility to infections with subsequent risk of liver failure, and patient death [10–12]. Nevertheless, the mechanism of neutrophil exhaustion has not been fully explained so far. Infections are still one of the leading causes of death in patients with end-stage liver disease. Therefore, both amelioration of liver function and prevention of infections are considered to be essential for ALD patient outcomes. Taking into account this background, we aimed to assess the systemic profile of neutrophil-derived mediators i.e. neutrophil elastase (NE), myeloperoxidase (MPO), as well as alpha1-antitrypsin (A1AT)- a potent inhibitor of neutrophil protease s, with an emphasis on their potential relevance in the course of ALD.

We aimed to answer the following questions:

1. What is the level of neutrophilic activation markers: alpha1-antitrypsin (A1AT), neutrophil elastase (NE), and myeloperoxidase (MPO) in the peripheral blood of ALD patients compared to the control group?

2. Are there differences in the level of A1AT, NE, and MPO in the blood depending on the sex of the patients?

3. Is there any correlation between the investigated markers (A1AT, NE, MPO) and the classical markers of an inflammatory response [CRP, WBC, neutrophil count, neutrophil-to-lymphocyte ratio (NLR)] in patients with ALD?

4. Is there a difference in the level of the aforementioned neutrophilic mediators depending on the liver function capacity assessed by criteria of the Child-Pugh classification (class A, B, C), MELD score ($<$20pts; $\geq$ 20pts), and the modified Maddrey discriminant function (mDF)?

5. Is there any correlation between the concentration of the molecules tested in the blood and the incidence of complications of ALD (i.e. ascites, liver encephalopathy, esophageal varices, hepatorenal syndrome, death)?

# 2. Materials and methods

## 2.1 Recruitment of study participants

The prospective study was conducted in the Department of Gastroenterology of the Medical University in Lublin, Poland. Adult inpatients (aged 18 years and over) with a diagnosis of

ALD were eligible to participate in the study. A priori approval was obtained from the institutional review board of the Medical University of Lublin (KE-0254/141/2010) which stated that our study protocol conforms to the ethical guidelines of the 1975 Declaration of Helsinki (6th revision, 2008).

According to the study protocol, patients signed informed consent and completed their medical history. A thorough interview was conducted with both a patient and their family members to confirm alcohol misuse problems in the year before the study. All patients answered the questions of the AUDIT-C (The Alcohol Use Disorders Identification Test—Consumption) questionnaire [13] https://www.hepatitis.va.gov/alcohol/treatment/audit-c.asp)) which is a modified version of the Alcohol Use Disorders Identification Test developed by the World Health Organization and published in 1998 [14]. The inclusion and exclusion criteria used during the study enrollment are listed below.

Inclusion criteria included:

1. Written consent of the patient to participate in the study;

2. Age eighteen years and above;

3. Data from the interview confirming alcohol abuse in amounts exceeding 40 g / day in men and 20 g / day in women during at least 6 months before study qualification;

4. Positive screening results obtained based on the AUDIT-C questionnaire (in men, a score of 4 or more, and in women, a score of 3 or more was considered a positive screen);

5. Common signs and symptoms of chronic liver disease on physical examination;

6. Laboratory test results consistent with alcoholic liver injury, this is elevated transaminases levels with the de Ritis ratio (AST/ALT ratio) above 2;

7. Exclusion of liver disease of other etiology (HBV and/or HCV infection, autoimmune hepatitis, Wilson's disease, hemochromatosis, drug-induced liver disease, cholestatic liver disease, etc).

Exclusion criteria included:

1. Lack of patient's written consent to participate in the study,

2. Previous treatment with corticosteroids and/or immunosuppressants in the last 6 months before study enrollment,

3. Chronic therapy with non-steroidal anti-inflammatory drugs and/or hepatotoxicity of other drugs noted in the last 6 months before study enrollment,

4. Transfusion of blood products during the last 6 months before study enrollment,

5. Coexistence of other serious diseases (eg cancer, circulatory failure, unstable diabetes mellitus, respiratory failure, chronic renal insufficiency, advanced coronary artery disease, hypothyroidism, hyperthyroidism, etc.).

## 2.2 Characteristics of the study cohort

Overall, 86 individuals were enrolled in the study including 62 patients with ALD, with or without signs of alcoholic hepatitis (AH) who created a study group, and 24 healthy volunteers from the control group matched to the study group in terms of age and sex. The demographic data of the study and control groups are presented in Table 1.

**Table 1. Demographic data of the study and control groups[a].**

|  | ALD group n = 62 | Controls n = 24 | p |
|---|---|---|---|
| **Age (years)** |  |  |  |
| Males | 45.84 ± 11.53 | 41.54 ± 9.82 | 0.07 |
| Females | 46.00 ± 13.09 | 45.38 ± 14.99 | 0.10 |
| **Gender n (%)** |  |  |  |
| Males | 53 (85.5) | 16 (66.7) |  |
| Females | 9 (14.1) | 8 (33.3) | 0.06 |

[a] ALD- alcohol-related liver disease, p- the level of significance.

Patients enrolled in the study maintained abstinence from alcohol for a minimum of 24 hours before taking blood samples for laboratory tests. None was treated with corticosteroids or pentoxifylline on admission. All patient data necessary for further analysis were recorded and the scheduled procedures were performed within 48 hours from the moment of admitting a patient to the hospital. Following the study protocol, first of all, patients' interviews, as well as the AUDIT-C questionnaire data, were collected. Then, a physical examination was performed and blood samples for chemistry and immunological examinations were collected.

Patients with an established diagnosis of ALD have been enrolled and divided into subgroups based on their gender, degree of liver dysfunction, and complications of ALD present on admission to the hospital such as ascites, hepatic encephalopathy, esophageal varices, renal dysfunction, and non-survival within the 90-day follow-up period. Overt hepatic encephalopathy (HE) was classified according to West-Heaven criteria [15]. Signs and symptoms of cholestasis were defined based on the recommendations of the European Association for the Study of the Liver (EASL) as alkaline phosphatase (ALP) activity above 1.5 times the upper limit of normal (ULN) and gamma glutamyltranspeptidase (GGT) activity of more than 3 times ULN [16]. In each case with signs of cholestasis, the titer of anti-mitochondrial antibodies (AMA) was determined to exclude primary biliary cholangitis (PBC). Moreover, each patient underwent an abdominal ultrasound examination (USG scanning) in order to eliminate other causes of cholestasis (e.g. gallstone disease) and/or to confirm the presence of ascites. An endoscopic assessment of the upper gastrointestinal tract (gastroscopy) was performed to determine the presence and size of the esophageal varices. The features of kidney dysfunction were confirmed by elevated serum creatinine levels above the upper limit of the normal range (i.e. above 1.3 mg / dL). If any further doubts existed, computed tomography, magnetic resonance imaging, and/or Duplex Doppler ultrasound scanning were performed.

After hospital discharge, studied patients were followed at least once a month at the Gastroenterology Clinic for the next 90 days and/or during each subsequent hospitalization if required.

The control group consisted of healthy volunteers recruited mostly from academics and trainees of the Gastroenterology Department of the Medical University of Lublin, who declared complete abstinence or alcohol consumption in an amount not exceeding 20 g/day.

As the current evidence indicates a different susceptibility to alcohol harm in women, the characteristics of the patients (Table 2) and the analysis of studied biomarker blood levels were checked in the subgroups of ALD patients allocated according to their gender. However, no statistically significant differences were observed based on patient gender.

**2.2.1 AUDIT-C questionnaire.** The AUDIT-C (Alcohol Use Disorders Identification Test—Consumption) questionnaire was used in order to identify persons with alcohol misuse who are hazardous drinkers. Its quality and usefulness have been confirmed in previous clinical trials [17, 18].

**Table 2. Characteristics of patients with ALD according to their gender distribution (Mann-Whitney test)[a].**

| | ALD study group n = 62 | | | | | | P |
|---|---|---|---|---|---|---|---|
| | Females with ALD n = 9 | | | Males with ALD n = 53 | | | |
| | median | 95% CI | 5–95 P | median | 95% CI | 5–95 P | |
| Age [years] | 56.00 | 43.10–58.86 | 26.00–61.00 | 47.00 | 43.76–52.00 | 33.00–64.00 | 0.27 |
| ALT IU/L | 38.00 | 30.00–76.83 | 22.00–480.00 | 42.00 | 34.76–62.71 | 17.50–228.25 | 0.93 |
| AST IU/L | 112.00 | 53.28–204.86 | 43.00–550.00 | 101.00 | 81.99–135.36 | 34.40–360.40 | 0.74 |
| ALP IU/L | 121.00 | 78.97–230.65 | 73.00–411.00 | 153.00 | 102.03–181.00 | 56.50–405.15 | 0.75 |
| GGT IU/L | 444.00 | 398.24–1610.32 | 155.00–2193.00 | 378.00 | 306.59–492.00 | 43.20–2558.20 | 0.16 |
| T-Bilirubin [mg/dL] | 1.70 | 0.65–18.74 | 0.60–34.10 | 2.90 | 1.90–5.55 | 0.60–16.57 | 0.82 |
| Albumin [g/dL] | 2.94 | 2.57–3.82 | 2.43–4.62 | 3.08 | 2.87–3.36 | 2.00–4.13 | 0.73 |
| INR | 1.33 | 1.00–2.19 | 0.60–2.54 | 1.29 | 1.21–1.42 | 0.93–2.24 | 0.87 |
| Creatynine [mg/dL] | 0.60 | 0.50–2.15 | 0.40–2.90 | 0.80 | 0.70–0.82 | 0.42–1.57 | 0.14 |
| CRP [mg/L] | 23.90 | 4.64–100.80 | 0.53–109.70 | 20.23 | 11.53–29.86 | 1.69–144.83 | 0.74 |
| WBC [x103cells/uL] | 5.79 | 4.54–9.08 | 4.07–13.55 | 7.00 | 5.56–8.19 | 3.25–15.95 | 0.94 |
| NEU [x103cells/uL] | 3.57 | 3.21–8.01 | 2.74–12.52 | 54.52 | 3.44–6.40 | 1.51–13.86 | 0.86 |
| LYM [x103cells/uL] | 0.92 | 0.52–1.51 | 0.46–1.55 | 1.06 | 0.96–1.32 | 0.45–2.37 | 0.39 |
| NLR | 4.03 | 2.27–13.21 | 2.06–27.22 | 3.98 | 3.17–6.06 | 1.47–14.81 | 0.68 |
| CTP | 8.00 | 6.00–12.86 | 5.00–13.00 | 8.00 | 7.00–9.00 | 5.00–13.00 | 0.78 |
| MELD | 11.00 | 8.00–28.00 | 7.00–39.00 | 15.00 | 11.00–17.24 | 6.25–25.00 | 0.89 |
| mDF | 15.40 | 9.49–89.16 | 1.06–102.44 | 24.40 | 11.40–35.41 | 2.21–75.39 | 0.88 |

[a] ALD- alcohol-related liver disease, ALT- alanine aminotransferase; AST- aspartate aminotransferase; CI- confidence interval, CRP-reactive protein; CTP- Child-Turcotte-Pugh score; ALP- alkaline phosphatase; GGT- gamma-glutamyl transpeptidase; INR- international normalized ratio; LYM- lymphocytes; mDF- modified Maddrey's discriminant function; MELD- Model of End-Stage Liver Disease score; NEU- neutrophils; NLR- neutrophil-to-lymphocyte ratio; P- percentile, p- the level of significance, T-bilirubin- total bilirubin; WBC- white blood cells.

The AUDIT-C questionnaire form (Table 3) was presented to the patients as follows:
One drink means:
1 can or bottle of beer,
1 glass of wine,
1 one cocktail,
1 glass of strong alcohol—e.g. vodka, gin, cognac.

**2.2.2 Determination of routine blood chemistry tests.** Routine laboratory tests were carried out in the Central Laboratory of the Independent Public Academic Hospital No. 4 in Lublin and included evaluations of:

**Table 3. AUDIT-C scoring.**

| | Score | | | | | Patient's points |
|---|---|---|---|---|---|---|
| Questions | 0 | 1 | 2 | 3 | 4 | |
| How often do you have a drink that contains alcohol? | Never | Monthly or less | 2–4 times per month | 2–3 times per week | 4 + times per week | |
| How many standard drinks of alcohol do you drink on a typical day when you are drinking? | 1–2 | 3–4 | 5–6 | 7–9 | 10 + | |
| How often do you have 5 or more drinks on one occasion? | Never | Less than monthly | Monthly | Weekly | Daily or almost daily | |
| **Total score** | | | | | | |

In men, a score of 4 or more, and in women, a score of 3 or more was considered positive.

- liver function parameters (alanine aminotransferase—ALT, aspartate aminotransferase—ASP, alkaline phosphatase—ALP, gamma-glutamyl transpeptidase- GGT, total bilirubin, albumin, prothrombin time (PT) with the international normalized ratio (INR);

- complete blood count (CBC) with White Blood Cell (WBC) Differential (hemoglobin level-Hgb; red blood cell- RBC count, platelet- PLT count, and white blood cell- WBC count, neutrophil- NEU count, and lymphocyte- LYM count),

- kidney function tests–creatinine and urea levels, serum sodium (Na) and potassium (K) levels,

- routine markers of inflammation such as WBC count and NEU count, neutrophil to lymphocyte ratio (NLR), and the level of C reactive protein (CRP),

- markers of other possible causes of chronic liver disease, i.e. HBV (HBs antigen, anti-HBc IgM, and IgG antibodies) and HCV (anti-HCV antibodies), anti-nuclear, anti-smooth muscle, and anti-mitochondrial antibodies, indicators of Wilson's disease and hemochromatosis.

Based on the obtained results, the degree of liver dysfunction was determined according to Child-Turcotte-Pugh (CTP) and Model of End-Stage Liver Disease (MELD) scores. Calculators available online [19, 20] were used for the assessment.

**2.2.3 Determination of serum neutrophilic mediator concentrations.** Four milliliters (ml) of peripheral blood were collected from the ulnar vein (EDTA tubes; Medlab, UK). Immediately after the collection, samples were centrifuged for 10 minutes at 3000 rpm. The serum obtained was aliquoted and stored at (-80)˚C until measurements were made.

The following commercially available enzyme-linked immunosorbent assays (ELISAs) were used for the determination of neutrophilic mediators:

1. Human Myeloperoxidase Quantikine ELISA Kit (R&D Systems);

2. Human Serpin A1 DuoSet ELISA (R&D Systems);

3. Neutrophil elastase, Human, ELISA kit (Hycult BIOTECH).

The examinations were carried out under the manufacturers' recommendations which were included in the protocols attached to the laboratory kits. Immunological tests of the studied patients and volunteers from the control group were performed at the Department of Immunology of the Medical University of Lublin (the head is Prof. Jacek Roliński), using the equipment available there.

**2.2.4 Statistical data analysis.** An assessment of the normality of data was performed using the Kolmogorov-Smirnow test. Due to the skewed values of the data set, the Mann-Whitney U test was used to compare the significance of differences between quantitative variables. Differences for qualitative variables were tested using the $\chi2$ test or Fisher's exact test for the small group size. Kruskal-Wallis and post hoc Dunn tests were used to check differences in biomarker levels of patients with different severity of liver dysfunction classified according to the CTP score. The relationships between the parameters of the liver function and the examined biomarkers were analyzed using Spearman's correlation test. The level of significance was set at a two-tailed $p < 0.05$.

# 3. Results

## 3.1 Comparison of serum neutrophilic biomarker concentrations in patients with ALD and individuals in the control group

The levels of alpha1-antitrypsin (A1AT), neutrophil elastase (NE), and myeloperoxidase (MPO) in the peripheral blood of ALD patients were assessed and compared to the control group. The obtained results showed a significantly higher concentration of neutrophilic

**Table 4. Comparison of serum A1AT, NE, and MPO concentrations in patients with ALD and volunteers in the control group (Mann-Whitney test)[a].**

| Biomarker [ng/mL] | ALD group (n = 62) | | Controls (n = 24) | | |
|---|---|---|---|---|---|
| | median | 5–95 P | median | 5–95 P | p |
| A1AT | 79.76 | 9.62–267.84 | 81.21 | 2.83–149.65 | 0.96 |
| NE | 240.18 | 104.53–514.22 | 107.22 | 51.80–341.24 | <0.0001* |
| MPO | 486.77 | 106.77–801.97 | 212.85 | 114.92–401.11 | <0.0001* |

[a] A1AT- alpha1-antitrypsin, ALD- alcohol-related liver disease; NE- neutrophil elastase, MPO- myeloperoxidase; P- percentile, p- the level of significance.

* statistically significant.

elastase (p<0.0001) and myeloperoxidase (p<0.0001) in the ALD group compared to controls. A1AT concentrations revealed no significant differences (p = 0.35) (Table 4; Fig 1).

## 3.2 Comparison of serum levels of neutrophilic biomarkers in subgroups of patients with ALD assigned by gender

Our analysis did not reveal significant differences in serum neutrophilic biomarker levels in ALD subgroups assigned by patient gender. The results are presented in Table 5 and Fig 2.

## 3.3 Analysis of correlations of serum neutrophilic activation biomarkers with laboratory parameters (i.e. liver tests, routine markers of inflammation) and Child-Turcotte-Pugh, MELD, and mDF scores

The potential correlations of the concentrations of the studied biomarkers with liver parameters and other lab tests were assessed.

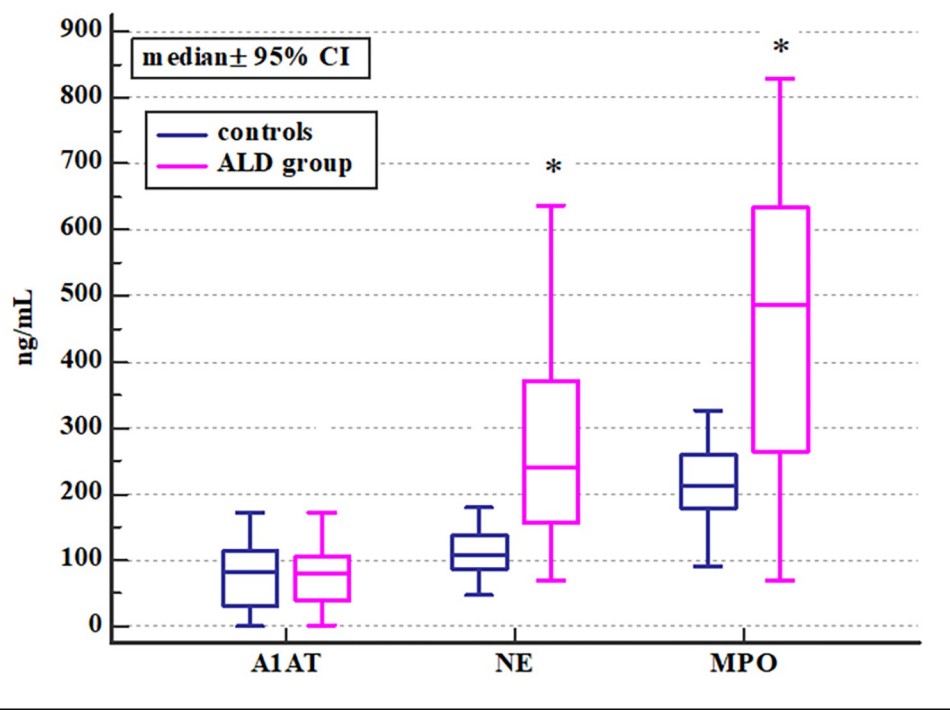

**Fig 1. Comparison of serum A1AT, NE, and MPO concentrations in patients with ALD and healthy volunteers in the control group (Mann-Whitney test)[a].** * p< 0.0001. [a] A1AT- alpha1-antitrypsin, ALD- alcohol-related liver disease, CI- confidence interval, NE- neutrophil elastase, MPO- myeloperoxidase.

**Table 5. Comparison of serum A1AT, NE, and MPO concentrations in men and women with ALD (Mann-Whitney test)[a].**

| Neutrophilic biomarker [ng/mL] | ALD males n = 53 | | ALD females n = 9 | | p |
|---|---|---|---|---|---|
| | median | 5–95 P | median | 5–95 P | |
| A1AT | 80.26 | 6.94–286.19 | 53.18 | 11.70–116.01 | 0.26 |
| NE | 232.31 | 102.13–488.67 | 276.34 | 186.33–560.47 | 0.27 |
| MPO | 410.50 | 86.32–808.83 | 577.66 | 192.04–791.47 | 0.29 |

[a] A1AT- alpha1-antitrypsin, ALD- alcohol-related liver disease; NE- neutrophil elastase, MPO- myeloperoxidase; P-percentile, p- the level of significance.

The negative correlation between albumin and MPO and a positive correlation between INR and NE were confirmed (Table 6, Figs 3 and 4).

Subsequently, we checked whether there was any relationship between the studied biomarkers (A1AT, NE, MPO) and routine markers of inflammation (i.e. CRP, WBC, neutrophils, neutrophils to lymphocytes ratio-NLR) in ALD patients. Positive correlations were found between MPO and CRP, WBC, and NEU, as well as NE and NEU, and NLR (Table 7).

We also tested if WBC differentials (NEU, LYM, neutrophil-to-lymphocyte ratio-NLR) influenced liver function parameters. Significant positive correlations were confirmed for AST and WBC, NEU, and NLR, as well as ALT and NLR (Table 8).

Further analysis of the results in patients with ALD revealed positive correlations between serum levels of NE and scoring of two scales i.e. MELD and mDF (Table 9).

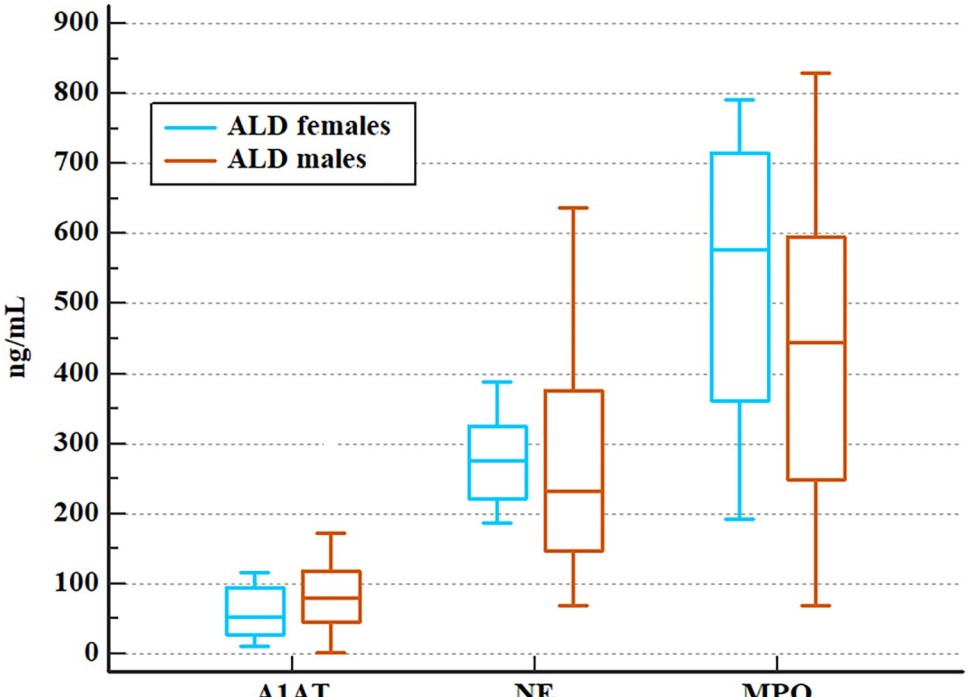

**Fig 2. Comparison of serum A1AT, NE, and MPO concentrations in men and women with ALD (Mann-Whitney test)[a].** Median ± 95% CI. [a] A1AT- alpha1-antitrypsin, ALD- alcohol-related liver disease; CI- confidence interval, NE- neutrophil elastase, MPO- myeloperoxidase.

**Table 6. Evaluation of the correlations of serum A1AT, MPO, and NE concentrations with liver parameters in patients with ALD (Spearman's rank correlation)[a].**

|  |  | A1AT | NE | MPO |
|---|---|---|---|---|
| ALT | Rho | -0.149 | 0.073 | 0.001 |
|  | P | 0.3285 | 0.5675 | 0.9962 |
| ASP | Rho | -0.203 | 0.149 | 0.034 |
|  | P | 0.1868 | 0.2444 | 0.8271 |
| ALP | Rho | 0.258 | -0.173 | -0.168 |
|  | P | 0.0992 | 0.1867 | 0.2865 |
| GGT | Rho | -0.025 | -0.019 | -0.130 |
|  | P | 0.8710 | 0.8816 | 0.3943 |
| T-bilirubin | Rho | 0.128 | 0.170 | -0.030 |
|  | P | 0.4029 | 0.1805 | 0.8429 |
| Albumin | Rho | -0.070 | -0.235 | -0.301 |
|  | P | 0.6682 | 0.0728 | 0.0492* |
| INR | Rho | 0.224 | 0.268 | -0.054 |
|  | P | 0.1395 | 0.0322* | 0.7245 |

[a] A1AT- alpha1-antitrypsin, ALD- alcohol-related liver disease; ALP- alkaline phosphatase, ALT- alanine aminotransferase, AST- aspartate aminotransferase; GGT- gamma-glutamyl transpeptidase; INR- international normalized ratio, NE- neutrophil elastase, MPO- myeloperoxidase, p- the level of significance, Rho- Spearman's coefficient of rank correlation, T-bilirubin- total bilirubin.

* statistically significant.

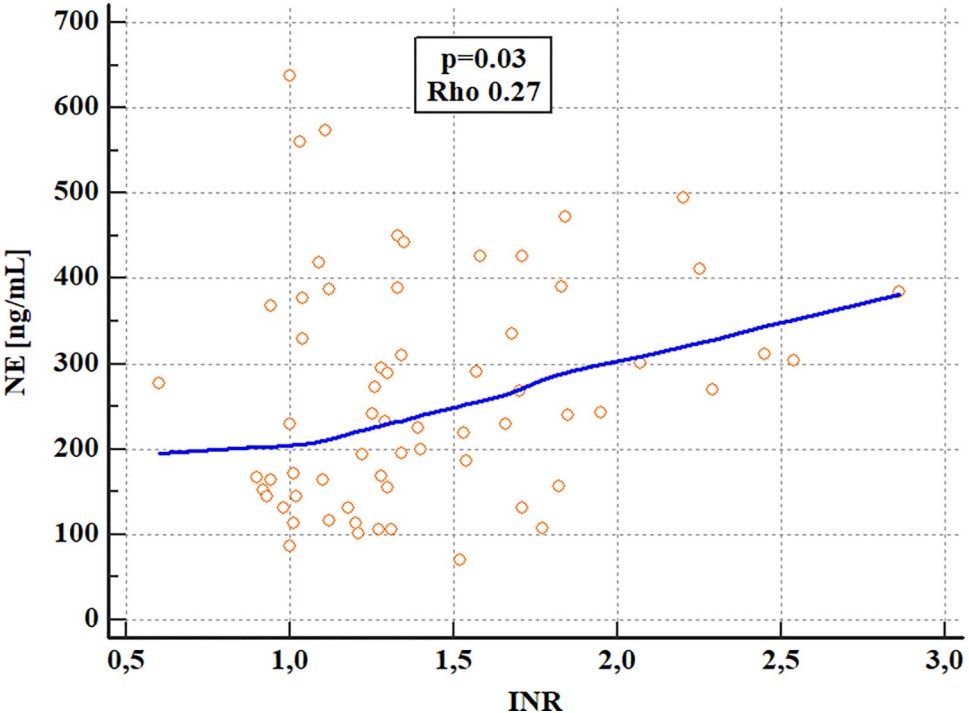

**Fig 3. Analysis of correlations of serum NE levels with INR in patients with ALD (Spearman's rank correlation)[a].** [a] ALD- alcohol-related liver disease, INR- international normalized ratio, NE- neutrophil elastase, p- the level of significance, Rho- Spearman's coefficient of rank correlation.

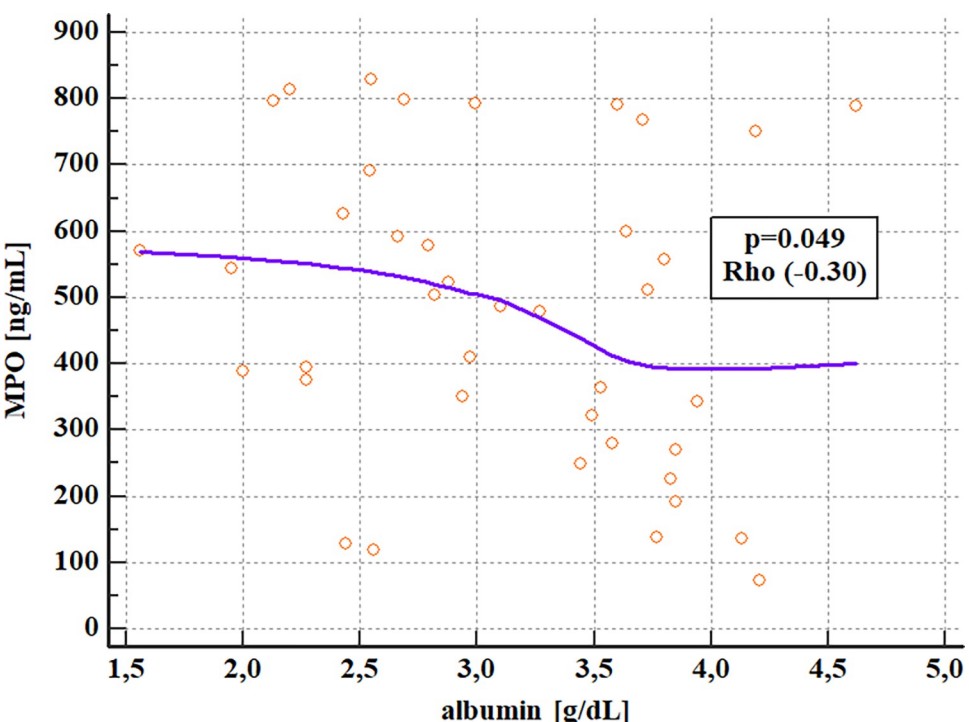

**Fig 4. Analysis of correlations of serum MPO and albumin levels in patients with ALD (Spearman's rank correlation)[a].** [a] ALD- alcohol-related liver disease, MPO- myeloperoxidase, p- the level of significance, Rho-Spearman's coefficient of rank correlation.

**Table 7. Analysis of correlations of serum A1AT, NE, and MPO concentrations and routine markers of inflammation in patients with ALD (Spearman's rank correlation)[a].**

|      |     | A1AT    | NE       | MPO      |
|------|-----|---------|----------|----------|
| CRP  | Rho | -0.006  | 0.173    | 0.320    |
|      | p   | 0.9681  | 0.1755   | 0.0342*  |
| WBC  | Rho | 0.160   | 0.200    | 0.319    |
|      | Pp  | 0.2943  | 0.1155   | 0.0328*  |
| NEU  | Rho | 0.130   | 0.252    | 0.348    |
|      | p   | 0.3933  | 0.0463*  | 0.0192*  |
| LYM  | Rho | 0.114   | -0.112   | 0.091    |
|      | p   | 0.4550  | 0.3843   | 0.5524   |
| NLR  | Rho | 0.022   | 0.308    | 0.286    |
|      | p   | 0.8839  | 0.0140*  | 0.0570   |

[a] A1AT- alpha1-antitrypsin, ALD- alcohol-related liver disease, CRP- C-reactive protein; LYM- lymphocytes; MPO-myeloperoxidase, NE- neutrophil elastase; NEU- neutrophils; NLR- neutrophil-to-lymphocyte ratio; p- the level of significance, Rho- Spearman's coefficient of rank correlation; WBC- white blood cells.

* statistically significant.

**Table 8. Analysis of correlations of WBC differentials and liver function parameters in patients with ALD (Spearman's rank correlation)[a].**

|  |  | AST | ALT | ALP | GGT | albumin | T-bilirubin | INR |
|---|---|---|---|---|---|---|---|---|
| **WBC** | Rho | 0.312 | 0.187 | 0.127 | 0.201 | -0.109 | 0.119 | 0.104 |
|  | p | 0.0136* | 0.1417 | 0.3369 | 0.1151 | 0.4170 | 0.3536 | 0.4174 |
| **NEU** | Rho | 0.335 | 0.200 | 0.070 | 0.215 | -0.085 | 0.103 | 0.108 |
|  | p | 0.0077* | 0.1158 | 0.6000 | 0.0907 | 0.5258 | 0.4226 | 0.3992 |
| **LYM** | Rho | -0.017 | -0.096 | 0.226 | 0.087 | -0.081 | -0.039 | -0.045 |
|  | p | 0.8964 | 0.4531 | 0.0848 | 0.4960 | 0.5447 | 0.764 | 0.7266 |
| **NLR** | Rho | 0.345 | 0.255 | -0.020 | 0.199 | -0.107 | 0.149 | 0.078 |
|  | p | 0.0060* | 0.0437* | 0.8831 | 0.1175 | 0.4229 | 0.2441 | 0.5438 |

[a] Rho- Spearman rank correlation coefficient; ALP- alkaline phosphatase; ALT- alanine aminotransferase; AST- aspartate aminotransferase; GGT- gamma-glutamyl transpeptidase; INR- international normalized ratio; LYM- lymphocytes; NEU- neutrophils; NLR- neutrophil-to-lymphocyte ratio; p- the level of significance, T-bilirubin- total bilirubin; WBC- white blood cells.

* statistically significant.

### 3.4 Comparison of serum neutrophilic biomarker concentrations in ALD patient subgroups allocated by different degrees of liver dysfunction classified according to Child-Turcotte-Pugh (CTP), Model of End-Stage Liver Disease (MELD) scores, and modified Maddrey's discriminant function (mDF)

Differences in serum levels of the biomarkers studied were evaluated based on the degree of liver dysfunction classified according to the CTP scale (class A, B, C), MELD (<20 points or ≥ 20 points), and mDF scores. There were no statistically significant differences in serum levels of A1AT, NE, and MPO in patients with ALD with different classes of CTP (Fig 5).

Patients with MELD scores greater than 20 points had significantly higher serum NE concentrations compared to patients with MELD < = 20 points. No statistically significant differences were found in terms of the levels of the remaining biomarkers (i.e. A1AT and MPO) in subgroups of patients with MELD scores above 20 points and MELD < = 20 points (Table 10).

Moreover, patients with severe alcoholic hepatitis with mDF>32 also had significantly higher serum NE concentrations compared to patients with mDF< = 32. No statistically significant differences were found in concentrations of the remaining biomarkers (i.e. A1AT and MPO) in mDF subgroups (Table 11).

**Table 9. Analysis of correlations of serum A1AT, NE, and MPO concentrations with the scoring systems of the severity of liver dysfunction in patients with ALD (Spearman's rank correlation)[a].**

| Scoring Scale |  | A1AT | NE | MPO |
|---|---|---|---|---|
| **CTP** | Rho | 0.006 | 0.214 | 0.195 |
|  | p | 0.9688 | 0.0901 | 0.2004 |
| **MELD** | Rho | 0.174 | 0.284 | 0.006 |
|  | p | 0.2538 | 0.0489* | 0.9698 |
| **mDF** | Rho | 0.127 | 0.303 | -0.005 |
|  | p | 0.4043 | 0.0150* | 0.9726 |

[a] A1AT- alpha1-antitrypsin, CTP- Child-Turcotte-Pugh score; NE- neutrophil elastase, mDF- modified Maddrey's discriminant function; MELD- Model of End-Stage Liver Disease score, MPO- myeloperoxidase; p- the level of significance, Rho- Spearman's coefficient of rank correlation.

* statistically significant.

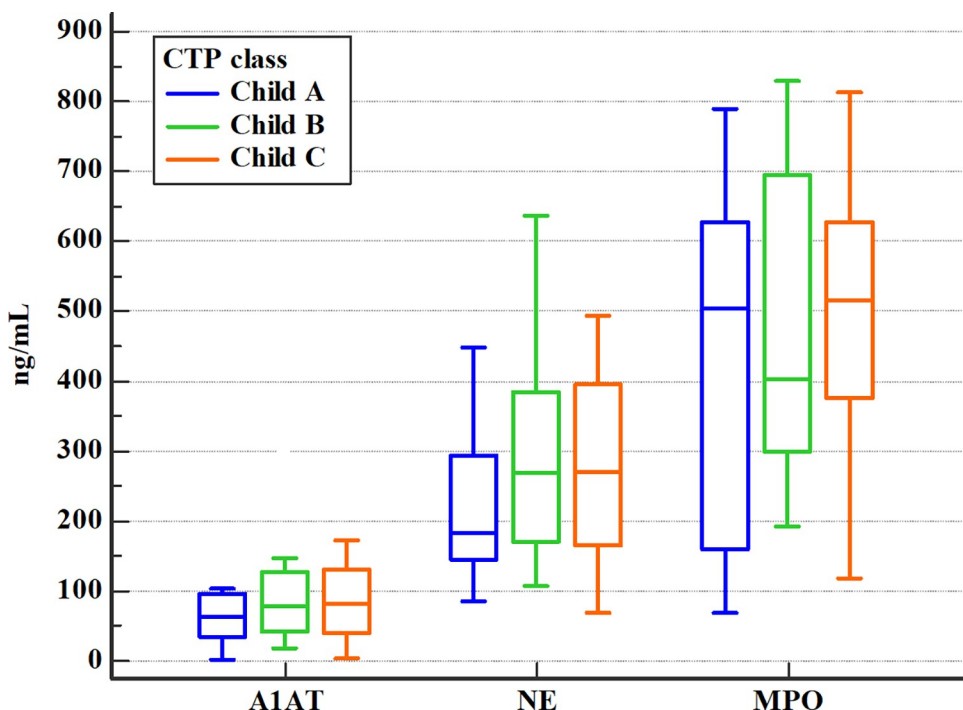

**Fig 5. Comparison of serum A1AT, NE, and MPO levels in patients with ALD with different CTP classes (Kruskal-Wallis test)[a].** [a] A1AT- alpha1-antitrypsin, ALD- alcohol-related liver disease, CTP- Child-Turcotte-Pugh score; NE- neutrophil elastase, MPO- myeloperoxidase.

Furthermore, we estimated the discriminative accuracy of NE concentrations for the assessment of ALD severity (i.e. mDF>32 and MELD>20) in comparison with routinely used lab parameters of the aforementioned scales. Receiver operating characteristic curves (ROC) were drawn and their Areas Under the ROC Curve (AUC) were compared. The results are presented in Tables 12 and 13, Figs 6 and 7.

Our results showed that the discriminative accuracy of NE concentrations for the assessment of ALD severity (i.e. mDF>32 and MELD>20) was lower than most routinely used lab parameters, but comparable to creatinine helpful for MELD score assessment.

## 3.5 Comparison of serum neutrophilic biomarker concentrations in subgroups of ALD patients allocated by end-stage liver disease complications

Serum levels of A1AT, NE, and MPO were also compared in subgroups of patients with ALD complications found on hospital admission. Our analysis revealed that there were no

**Table 10. Comparison of serum A1AT, NE, and MPO concentrations in patients with ALD assigned by liver dysfunction according to the MELD score (Mann-Whitney test)[a].**

| Biomarker ng/mL | ALD patients with MELD>20 (n = 14) | | ALD patients with MELD < = 20 (n = 48) | | p |
|---|---|---|---|---|---|
| | median | 5–95 P | median | 5–95 P | |
| A1AT | 94.38 | 11.88–319.82 | 73.99 | 6.47–235.40 | 0.27 |
| NE | 310.83 | 106.87–483.87 | 229.50 | 106.12–560.47 | 0.037* |
| MPO | 533.48 | 123.54–812.54 | 486.77 | 83.99–797.23 | 0.66 |

[a] A1AT- alpha1-antitrypsin, ALD- alcohol-related liver disease; NE- neutrophil elastase, MPO- myeloperoxidase; P-percentile, p- the level of significance.
* statistically significant.

**Table 11. Comparison of serum A1AT, NE, and MPO levels in patients with ALD based on the degree of alcoholic hepatitis and mDF scores (Mann-Whitney test)[a].**

| Biomarker ng/mL | ALD patients with mDF>32 n = 28 | | ALD patients with mDF< = 32 n = 34 | | p |
|---|---|---|---|---|---|
| | Median | 5–95 P | Median | 5–95 P | |
| A1AT | 80.78 | 2.56–242.21 | 73.99 | 19.95–269.88 | 0.78 |
| NE | 292.14 | 102.13–474.92 | 194.46 | 102.67–568.71 | 0.030* |
| MPO | 394.65 | 96.11–820.55 | 507.75 | 116.94–794.92 | 0.57 |

[a] A1AT- alpha1-antitrypsin, ALD- alcohol-related liver disease; NE- neutrophil elastase, mDF MPO- myeloperoxidase; P-percentile, p- the level of significance.

* statistically significant.

**Table 12. Estimation of NE discriminative accuracy for mDF>32 in patients with ALD in comparison with routinely used lab parameters.**

| Variable | AUC | SE [a] | 95% CI [b] |
|---|---|---|---|
| NE | 0.659 | 0.0706 | 0.530 to 0.773 |
| T-Bilirubin | 0.844 | 0.0503 | 0.732 to 0.923 |
| INR | 0.986 | 0.00997 | 0.919 to 1.000 |

[a] DeLong et al., 1988

[b] Binomial exact.

AUC- Area Under the ROC Curve; CI- confidence interval; INR- international normalized ratio, NE- neutrophil elastase, SE- standard error.

**Table 13. Estimation of NE discriminative accuracy for MELD>20 in patients with ALD in comparison with routinely used lab parameters.**

| Variable | AUC | SE [a] | 95% CI [b] |
|---|---|---|---|
| NE | 0.679 | 0.0800 | 0.550 to 0.790 |
| T-bilirubin | 0.904 | 0.0521 | 0.804 to 0.963 |
| INR | 0.886 | 0.0480 | 0.782 to 0.952 |
| Creatinine | 0.659 | 0.0843 | 0.530 to 0.773 |

[a] DeLong et al., 1988

[b] Binomial exact/

ALD- alcohol-related liver disease, AUC- Area Under the ROC Curve; CI- confidence interval; INR- international normalized ratio, MELD- Model of End Stage Liver Disease, NE- neutrophil elastase, SE- standard error, T-bilirubin-total bilirubin.

significant differences in the levels of studied neutrophilic biomarkers in patients with or without ALD complications (i.e. ascites, hepatic encephalopathy-HE, esophageal varices-EV, renal dysfunction, survival). The results are presented in Table 14.

## 4. Discussion

Alcohol-related liver disease (ALD) remains the leading cause of chronic liver failure worldwide [21]. New concepts of liver damage based on results from recent clinical and experimental trials of animal and cell culture models indicate that neutrophils, as the most abundant type of granulocytes and key immune players, are an important pathogenetic link in the development of liver diseases, regardless of etiology, including ALD [22]. Ethanol-induced damage

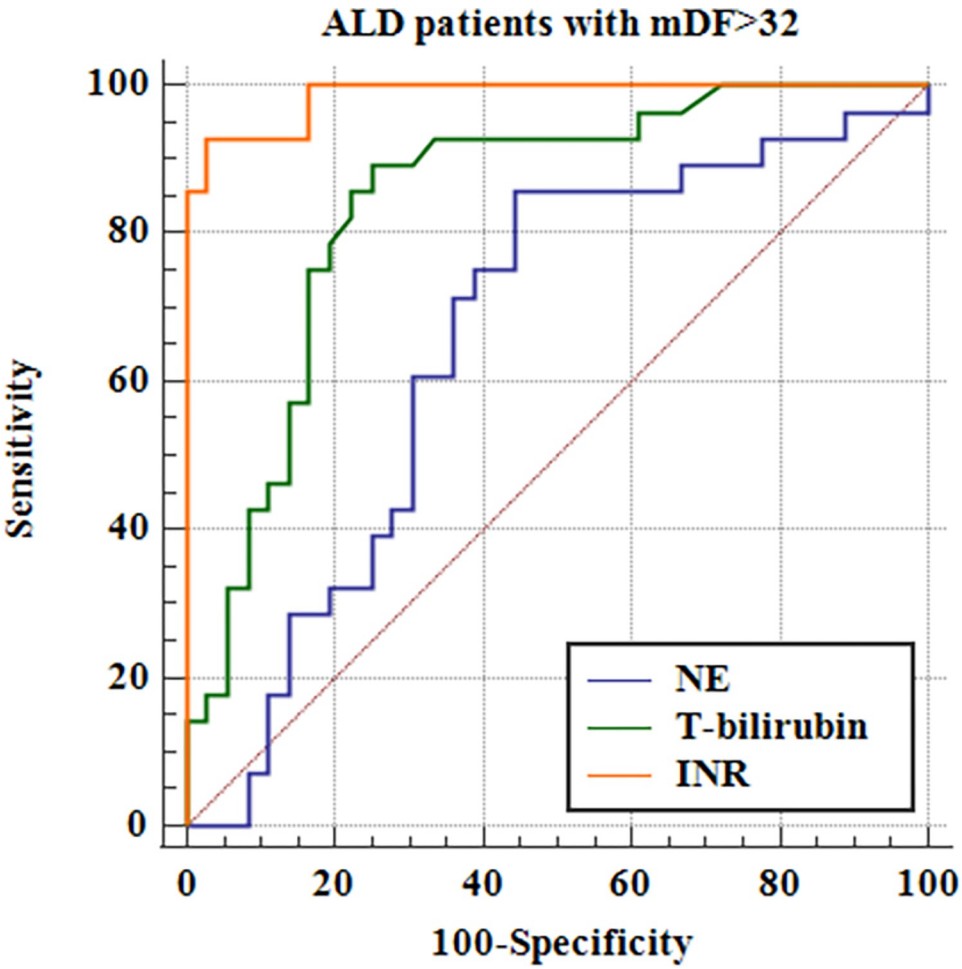

**Fig 6. Evaluation of NE discriminative accuracy for mDF>32 in patients with ALD in comparison with routinely used lab parameters[a]. [a]** ALD- alcohol-related liver disease, NE- neutrophil elastase, INR- international normalized ratio, mDF- modified Maddrey's discriminant function, T-bilirubin- total bilirubin.

and inflammation in liver parenchyma stimulate increased neutrophil recruitment and activation [8, 9]. For several years, these cells were considered simply micro-organism "killers" during the immune response. However, recent studies have revealed more advanced mechanisms related to their cytotoxic function and their potential implications in the pathogenesis of various infectious and inflammatory diseases [23–25]. Neutrophil chemotaxis to inflammation/injury sites is mediated by their interaction with membrane cytokine and/or chemokine receptors and is critical in the hepatic inflammatory response [7, 22]. Neutrophils migrate from the liver sinusoids through the endothelium and penetrate the liver parenchyma, triggering an inflammatory response, hepatocyte damage, and organ fibrosis. They form the body's first line of defense, which is induced much faster than the acquired immune response [26, 27]. In the liver, the innate immune response is mediated by various subpopulations of white blood cells, including natural killers (NK), natural T-killers (NKT), dendritic cells (DC), neutrophils, eosinophils, and antimicrobial proteins. The most abundant neutrophils play an important role in the defense of our body against infections and sterile inflammation. [22, 28, 29]. Activation of the innate immune response by alcohol misuse with subsequent induction of hepatitis is mediated by ethanol-related translocation of liposaccharides from the intestinal lumen to

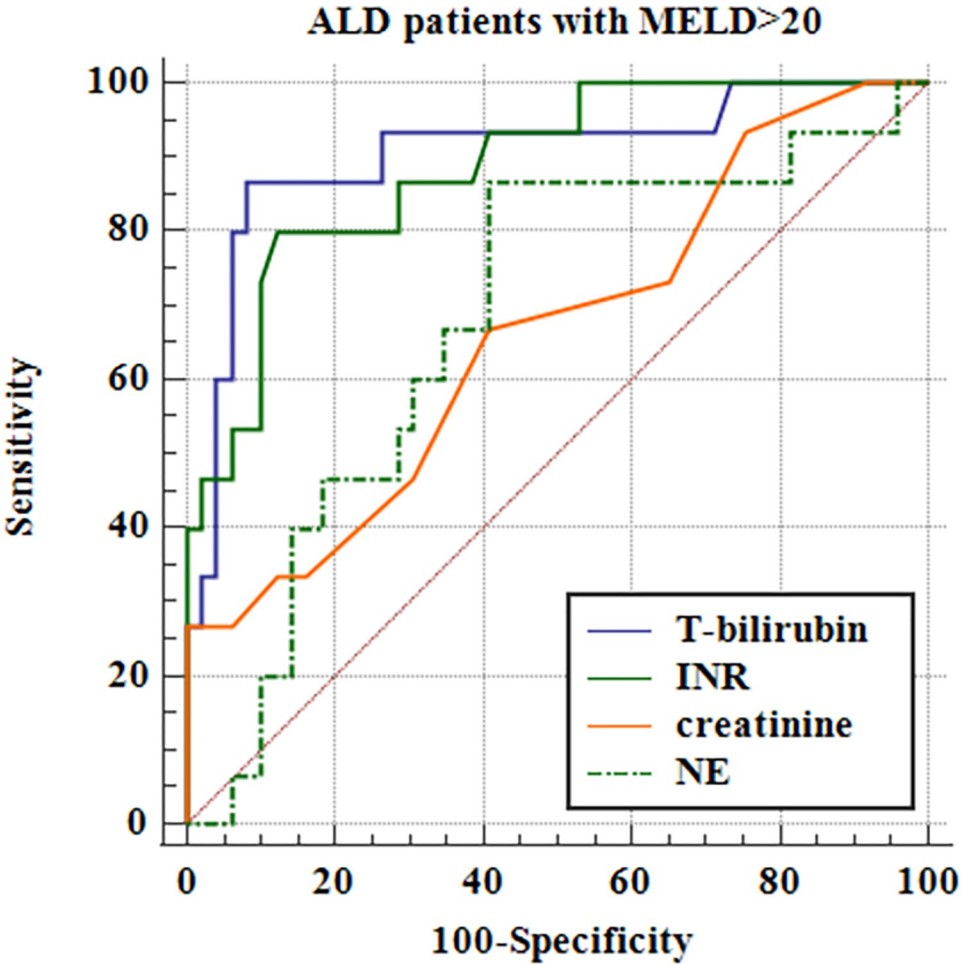

**Fig 7. Evaluation of NE discriminative accuracy for MELD>20 in patients with ALD in comparison with routinely used lab parameters[a].** [a] ALD- alcohol-related liver disease, INR- international normalized ratio, MELD- Model of End Stage Liver Disease, NE- neutrophil elastase, T-bilirubin- total bilirubin.

the portal circulation, further activation of Kupffer cells with increased release of pro-inflammatory cytokines and chemokines, and massive recruitment of neutrophils from peripheral blood to the liver [27, 28, 30]. Excessive neutrophil activation may lead to a so-called cytokine storm with the progressive destruction of the liver parenchyma. For this reason, it is considered a double-edged sword.

In our study, we investigated the systemic concentrations of three mediators released from activated neutrophils in patients with confirmed alcohol abuse. Their potential associations with the severity of the systemic inflammatory response, liver dysfunction, and the rate of complications of liver disease were also analyzed.

At the beginning of the study, serum levels of studied neutrophilic mediators i.e. alpha1-antitrypsin (A1AT), neutrophil elastase (NE), and myeloperoxidase (MPO) were compared in ALD patients and healthy volunteers from the control group. The results revealed that patients with ALD have significantly higher systemic concentrations of NE and MPO as compared to controls. This supports the concept of the involvement of neutrophil activation and their mediators in the pathogenesis of ALD. Since research indicates that women are more prone to alcohol-related liver damage and have a poorer prognosis [31], A1AT, NE, and MPO were

**Table 14. Comparison of serum A1AT, NE, and MPO concentrations in ALD patients with or without ALD complications (Mann-Whitney test)[a].**

| Biomarker [ng/mL] | Type of ALD complication | | | | p |
|---|---|---|---|---|---|
| | Ascites (+) n = 30 | | Ascites (-) n = 32 | | |
| | median | 5–95 P | median | 5–95 P | |
| A1AT | 80.77 | 7.53–313.00 | 68.51 | 15.69–193.68 | 0.94 |
| NE | 242.31 | 106.68–470.36 | 225.55 | 101.60–550.56 | 0.57 |
| MPO | 482.76 | 128.47–813.71 | 503.95 | 71.81–800.38 | 0.65 |
| | HE (+) n = 14 | | HE (-) n = 48 | | |
| | median | 5–95 P | median | 5–95 P | p |
| A1AT | 80.27 | 4.20–308.62 | 79.48 | 19.12–244.86 | 0.94 |
| NE | 279.64 | 113.04–476.26 | 229.52 | 97.79–563.01 | 0.28 |
| MPO | 506.31 | 237.85–827.68 | 410.50 | 79.50 to 795.66 | 0.76 |
| | EV (+) n = 28 | | EV (-) n = 34 | | |
| | median | 5–95 P | median | 5–95 P | p |
| A1AT | 87.83 | 7.12–203.78 | 68.51 | 14.56–305.71 | 0.99 |
| NE | 230.90 | 97.67–453.59 | 267.92 | 106.18–569.97 | 0.37 |
| MPO | 478.76 | 91.22–822.13 | 503.95 | 120.10–794.16 | 0.34 |
| | Renal dysfunction (+) n = 4 | | Renal dysfunction (-) n = 58 | | |
| | median | 5–95 P | median | 5–95 P | p |
| A1AT | 87.70 | 11.70–137.51 | 79.48 | 9.91–276.00 | 0.90 |
| NE | 279.64 | 171.45–494.40 | 241.68 | 102.93–524.88 | 0.28 |
| MPO | 634.01 | 226.14–813.41 | 478.76 | 97.68–796.99 | 0.26 |
| | Non-survival n = 3 | | Survival n = 59 | | |
| | median | 5–95 P | median | 5–95 P | p |
| A1AT | 64.23 | 15.27–80.78 | 80.26 | 8.37–273.96 | 0.32 |
| NE | 272.31 | 269.91–389.48 | 232.31 | 103.73–524.13 | 0.36 |
| MPO | 543.35 | 135.91–690.37 | 482.76 | 99.96–804.26 | 0.56 |

[a] A1AT- alpha1-antitrypsin, ALD- alcohol-related liver disease; EV- esophageal varices, HE- hepatic encephalopathy, NE- neutrophil elastase, MPO- myeloperoxidase; P-percentile, p- the level of significance.

compared in both sex subgroups. However, our analysis revealed no significant differences in serum levels of studied biomarkers in patient subgroups allocated by gender. Females and males with ALD had comparable concentrations of the three neutrophilic biomarkers.

A recent report suggests that aging worsens the course of ALD and fibrosis in mice by downgrading the expression of sirtuin 1 [32]. However, there were no correlations between A1AT, MPO, and NE concentrations and patient age in our cohort.

Some evidence indicates that corticosteroids provide a short-term survival benefit in about 50% of treated patients with severe alcoholic hepatitis (AH) [33]. Since white blood cell (WBC), neutrophil (NEU), lymphocyte (LYM) counts, and neutrophil-lymphocyte ratio (NLR) have been associated with disease severity, therefore we investigated their potential relationship with liver function parameters. Our results confirmed significant positive correlations of WBC, NEU count, and NLR with AST, as well as NLR with ALT. NLR can be easily obtained from simple blood tests. As a noninvasive biomarker of the systemic inflammatory response, it has been shown to independently predict the clinical outcome of various benign and malignant disorders [34–36].

As reported by Biyik et al. [37] in their retrospective observational cohort study, NLR was a predictor of mortality independent of CTP and MELD scores in patients with liver cirrhosis. It was able to predict patient non-survival also in subgroups with low MELD scores. Also, Zhang

et al. [38] observed that NLR was an independent predictor of 30-day and 90-day mortality in patients undergoing transjugular intrahepatic portosystemic shunt (TIPS). Moreover, in patients with MELD ≤ 15, it was a better prognostic factor compared to MELD or MELD-Na scores for the short-term outcome. Forrest et al. [39] also reported that the inclusion of the NLR index in the modified Glasgow Alcoholic Hepatitis Score (mGAHS) improved its AUC value for 28- and 90-day mortality in patients with AH.

Subsequently, we checked the potential association of the studied neutrophil biomarkers with the aforementioned routine inflammation markers and observed significant positive correlations of serum NE levels with NEU counts and NLR, as well as serum MPO levels with CRP, WBC, and NEU counts. Fujimoto et al. [40] showed that endotoxemia and elevated levels of IL-8 might play a key role in neutrophil activation and migration in patients with AH. Moreover, increased secretion of pro-inflammatory cytokines (IL-6 and IL-8) was associated with inflammatory exacerbation and poor patient prognosis in AH. Recent reports indicate that inflammatory mediators such as tumor necrosis factor (TNF), IL-1, platelet-activating factor (PAF), IL-8, HMGB-1 (high mobility group box protein 1), lipid peroxidation products released from dying or dead hepatocytes, and CXC chemokines are strong promoters of neutrophil extravasation in the hepatic tissue [41, 42]. Ziol et al. [43] analyzed the histopathological results of hepatic samples in 35 patients with AH and found that the hepatocyte apoptotic index and the neutrophil infiltration index were strongly correlated.

Our results are consistent with the above-described reports. We observed the positive correlation of NE concentrations with liver disease prognostic systems i.e. MELD and mDF scores. Serum NE levels were also significantly higher in patients with end-stage liver dysfunction defined as MELD> 20 points and severe AH with mDF>32. Nevertheless, the discriminative accuracy of NE concentrations for the assessment of ALD severity was lower than routinely used lab parameters except for creatinine (Tables 12 and 13; Figs 6 and 7). Moreover, serum NE concentrations revealed positive correlations with INR, while MPO had negative correlations with blood albumin levels. Both lab parameters are hepatic synthetic function indicators. Nevertheless, no differences were observed in serum levels of studied neutrophilic biomarkers in ALD patients allocated by liver disease complications and survival.

Recent evidence indicates that even in stable liver cirrhosis, the release of neutrophilic inflammatory mediators occurs in hepatic tissue [12]. Clinical observations show that despite the neutrophilic activation, paradoxically, microbial infections remain an important cause of death in ALD patients [44–46]. Under physiological conditions, MPO and the production of reactive oxygen species (ROS) by neutrophilic nicotinamide adenine dinucleotide phosphate (NADPH) oxidase lead to efficient microbial elimination [47]. The development of liver cirrhosis impairs immune function and increases susceptibility to infections. Boussif et al. 2016 [48] showed that neutrophils from patients with decompensated alcohol-related liver cirrhosis presented impaired MPO release and decreased bactericidal activity after stimulation with fPR by the formylpeptide formyl-met-leu-phe (fMLP).

The studies by Mookerjee et al. [49] confirmed that increased endotoxin activation of neutrophils decreases their ability to respond to other bacterial infections. They showed that patients with alcohol-related liver cirrhosis presented with immune cell dysfunction, which was significantly associated with the increased risk of disease complications such as infections, multiorgan failure, and mortality. However, neutrophil dysfunction was reversible after endotoxin removal from the blood. The results of those studies indicate the significant prognostic role of neutrophils in the pathogenesis of ALD. We are tempted to speculate that the assessment of neutrophil function might be a useful tool in the future selection of ALD patients for immune therapy.

In the last decade, numerous efforts have been made to determine the prognostic value of various biomarkers obtained from body fluids in patients with liver cirrhosis. Several reports indicate that neutrophil gelatinase-associated lipocalin (NGAL) could be a new and sensitive biomarker of tubular kidney injury in liver cirrhosis [50–52]. Fluid sampling is easy, non-invasive, and reproducible; therefore, such biomarkers could be ideal diagnostic and prognostic indicators. Accurate assessment of patient prognosis and the precise management of ALD are key factors in reducing the mortality rate. Currently, liver biopsy and measurements of the liver venous pressure gradient (HVPG) are considered the best available methods for determining the severity of liver disease [53]. However, their use is limited due to their invasiveness. Moreover, CTP and MELD systems are widely used for the prediction of disease outcomes, but their disadvantages, due to the subjectivity of the CTP score and the insufficient MELD scoring for all cirrhotic stages, reduce the chances of proper medical decision-making [54–56].

Our results should be interpreted with caution in the context of potential study limitations. Firstly, this was a single-center study with a relatively small sample size. Secondly, we have not evaluated the changes in neutrophilic biomarker levels in the same patient over time, therefore, treatment effects cannot be assessed concerning their serum concentrations. Third, since the present study was not designed to predict the ALD-related mortality rate, the relatively small number of nonsurvivors (only 3 persons) makes the prognostic evaluation impossible. Fourth, since patient alcohol consumption was self-reported, its true intake assessment may lack precision. Moreover, the impact of recall bias and/or intentional false reporting cannot be ruled out.

## 5. Conclusions

Our results point out the key role of neutrophils in the pathogenesis of ALD and support the value of NE and MPO in the prognostic evaluation of ALD patients. Both aforementioned neutrophilic biomarkers might be inflammatory indicators. Moreover, NE is the likely marker of disease severity. The high systemic NE/A1AT ratio may facilitate the spread of the inflammatory cascade in ALD. Our data showed that increased neutrophilic activation could be a trigger for ALD progression and unfavorable patient prognosis. Therefore, the NE and MPO blood concentration assessment seems to be useful for further potential inhibition of hepatic inflammation and fibrosis in an attempt to prevent the disease progression. Since no specific treatment for ALD has been discovered so far, immune modulation is under extensive investigation. Accurate targeting factors modifying neutrophil function might create a new direction for the treatment of ALD in the future. However, results of randomized clinical trials are eagerly awaited before the final adoption and incorporation of modern therapeutic modalities into routine clinical practice. As for the diagnostic approach, peripheral blood biomarkers are readily available, non-invasive, and do not require large financial outlays. As a result, they might be useful tools also in the long-term surveillance of liver transplant candidates. This justifies the need for further research in this area in larger groups of patients in multicenter trials.

## Author Contributions

**Conceptualization:** Beata Kasztelan-Szczerbinska, Jacek Rolinski, Halina Cichoz-Lach.

**Data curation:** Bartosz Zygo, Anna Rycyk-Bojarzynska, Agata Surdacka.

**Formal analysis:** Beata Kasztelan-Szczerbinska.

**Funding acquisition:** Beata Kasztelan-Szczerbinska, Halina Cichoz-Lach.

**Investigation:** Bartosz Zygo, Anna Rycyk-Bojarzynska.

**Methodology:** Bartosz Zygo, Agata Surdacka.

**Resources:** Beata Kasztelan-Szczerbinska, Halina Cichoz-Lach.

**Software:** Beata Kasztelan-Szczerbinska.

**Supervision:** Beata Kasztelan-Szczerbinska, Jacek Rolinski, Halina Cichoz-Lach.

**Validation:** Beata Kasztelan-Szczerbinska, Agata Surdacka.

**Visualization:** Beata Kasztelan-Szczerbinska.

**Writing – original draft:** Beata Kasztelan-Szczerbinska, Bartosz Zygo, Anna Rycyk-Bojarzynska.

**Writing – review & editing:** Beata Kasztelan-Szczerbinska, Jacek Rolinski, Halina Cichoz-Lach.

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
