## [Decision Letter · Decision Letter 0]

1 Dec 2022

PONE-D-22-28565Blood concentrations of mediators released from activated neutrophils are related to the severity of alcohol-induced liver damage. --PLOS ONE

Dear Dr. Kasztelan-Szczerbińska,

Thank you for submitting your manuscript to PLOS ONE. After careful consideration, we feel that it has merit but does not fully meet PLOS ONE’s publication criteria as it currently stands. Therefore, we invite you to submit a revised version of the manuscript that addresses the points raised during the review process.

ACADEMIC EDITOR: Please refer to the comments and revise your paper according to the comments of the reviewers.==============================

We look forward to receiving your revised manuscript.

Kind regards,

Gulali Aktas

Academic Editor

PLOS ONE

Journal Requirements:

3. We note you have included a table to which you do not refer in the text of your manuscript. Please ensure that you refer to Table 3 in your text; if accepted, production will need this reference to link the reader to the Table.

Additional Editor Comments:

There are several issues that must be revised before reconsideration of the paper. Please refer to reviewers' comments.

Reviewers' comments:

Reviewer's Responses to Questions

**Comments to the Author**

1. Is the manuscript technically sound, and do the data support the conclusions?

Reviewer #1: Partly

Reviewer #2: Yes

2. Has the statistical analysis been performed appropriately and rigorously? 

Reviewer #1: Yes

Reviewer #2: Yes

3. Have the authors made all data underlying the findings in their manuscript fully available?

Reviewer #1: Yes

Reviewer #2: Yes

4. Is the manuscript presented in an intelligible fashion and written in standard English?

Reviewer #1: Yes

Reviewer #2: Yes

5. Review Comments to the Author

Reviewer #1: Abstract

It is not usual to use the name "conventional" markers of inflammation. Please replace with a more adequate term. NE was related to MELD and mDF scores. What do you mean by this?

Keywords?

Blood biomarkers. Broad term, should be replaced

Introduction

Very clear and concise. However, this is the first time I've encountered a goal like this. He is really clear, however, the questions asked are not usual for the original article. If this is the policy of the journal, then it is ok. If not, it is necessary to modify the questions in, for example, aim: 2. Ln 86-87. Conducted study... This should be transferred to Material and methods

Material and methods

Ln 139 Please check the names of the parameters and add the correct abbreviations or description of the abbreviations where they are missing. For example: alanine aminotransferase is not A1AT, but ALT. Guanosine-5'-triphosphate is GTP. In your case it is gamma-glutamyl transpeptidase. Do you mean transaminotrasferase (GGT)? Why did you mention alkaline phosphatase in the assessment of liver function, when it is an organospecific enzyme for bones?

What is INR?

Ln, 141 it is necessary to equalize the WBC count and RBC count.

Abbreviations are missing again.

Ln 145 Replace the word conventional.

I believe that the methods should be classified into several groups. Please form adequate titles and subtitles. You can't write together about blood and serum parameters, then there is a division of patients, certain criteria and symptoms….

Table 2. GTP, please correct this cardinal error. Here it is now ALT, and in methods A1AT. What does the mark kom/uL represent? If the international normalized ratio (INR) is the unit for prothrombin time, there cannot be an INR in the table. It's the same for you if you wrote mmol/L instead of the glucose value. Ln instead of 700 x g specify rpm.

Table 4. Please enter a numerical value for each p value, and not combine p and significant levels. Place a mark next to the p value and mark the level of significance (p<0.000) under the table.

Tables 5 and 6 are not described. Remove n from Table 6 to make the table smaller.

Please eliminate the green coloring and mark significant differences with *.

The description of the images is also not given. It looks like you presented the data as figures next to the table.

The figures are not described at all. It is necessary to standardize all tables. It is better to eliminate the green ones, put a sign * and write an explanation for significance under the table. In general, a lot of data in the results is displayed chaotically.

3.5. There are no data.

Discussion

The first page of the discussion (ln 359-385) is a general story. This could have been the Introduction. The discussion is well written.

Reviewer #2: Thanks so much for trust me to review this article

I have just one recommendation to the authors is :

the type of the study must mention in the abstract

6. PLOS authors have the option to publish the peer review history of their article (what does this mean?). If published, this will include your full peer review and any attached files.

Reviewer #1: No

Reviewer #2: **Yes: **My name : Dr.Zubaida Falih Alzubaidi

---

## [Author Response · Author response to Decision Letter 0]

4 Dec 2022

Our rebuttal letter,

Manuscript PONE-D-22-28565, entitled: Blood concentrations of mediators released from activated neutrophils are related to the severity of alcohol-induced liver damage

We would like to thank both Reviewers for allowing us to improve our paper and submit a revised draft of our manuscript titled: Blood concentrations of mediators released from activated neutrophils are related to the severity of alcohol-induced liver damage, to Plos One. We appreciate the time and effort that the reviewers have dedicated to providing their valuable feedback and comments on our manuscript. We have incorporated changes reflecting all the suggestions advisable by the reviewers. The changes have been highlighted in yellow within the manuscript. 

Our response to Reviewer #1: (in blue)

Comment from Reviewer #1 

„Abstract

It is not usual to use the name "conventional" markers of inflammation. Please replace with a more adequate term.” 

Our response: We change to routine markers instead of using conventional markers, although the term conventional is widely used by researchers, for example: 

1. Jiang et al. Novel Surrogate Markers of CNS Inflammation in CSF in the Diagnosis of Autoimmune Encephalitis. Front. Neurol. 2020,10:1390. doi: 10.3389/fneur.2019.01390 „Supportive findings indicating AE include the following conventional surrogate markers of neuroinflammation on CSF: oligoclonal bands, raised protein level, and monocytosis”;

2. Guz G, Colak B, Hizel K, et al. Procalcitonin and Conventional Markers of Inflammation in Peritoneal Dialysis Patients and Peritonitis. Peritoneal Dialysis International. 2006;26(2):240-248. doi:10.1177/089686080602600221 

Comment from Reviewer #1 „NE was related to MELD and mDF scores. What do you mean by this?” 

Our response: Thank you for pointing this out. The sentence was corrected as follows- NE correlated with MELD and mDF scores.

Comment from Reviewer #1 „Keywords? Blood biomarkers. Broad term, should be replaced 

Our response: We have replaced the term with neutrophil-derived biomarkers.

Comment from Reviewer #1 „Introduction Very clear and concise. However, this is the first time I've encountered a goal like this. He is really clear, however, the questions asked are not usual for the original article. If this is the policy of the journal, then it is ok. If not, it is necessary to modify the questions in, for example, aim: 2. 

Our response: Yes, we tried to be clear and concise, therefore we presented all the details of our research in the simplest possible way starting from questions that we were going to answer by conducting our study. 

Comment from Reviewer #1 „Ln 86-87. Conducted study... This should be transferred to Material and methods”

Our response: The sentence has been transferred to Material and methods, as recommended by the Reviewer.

Comment from Reviewer #1 „Material and methods

Ln 139 Please check the names of the parameters and add the correct abbreviations or description of the abbreviations where they are missing. For example: alanine aminotransferase is not A1AT, but ALT. Guanosine-5'-triphosphate is GTP. In your case it is gamma-glutamyl transpeptidase. Do you mean transaminotrasferase (GGT)?”

Our response: We apologize for the mistakes. All of them have been corrected as follows: AST/ALT ratio; alanine aminotransferase – ALT; gamma-glutamyl transpeptidase- GGT. 

Comment from Reviewer #1 „Why did you mention alkaline phosphatase in the assessment of liver function, when it is an organospecific enzyme for bones?” 

Our response: While we appreciate the Reviewer’s feedback, we respectfully disagree. The two main blood sources of alkaline phosphatase (ALP) include the liver and bones. High levels of ALP may indicate bile duct disorders and should be checked together with GGT to confirm the presence of cholestasis. As we and other researchers have confirmed in previous studies, a cholestatic pattern with elevated ALP levels and intrahepatic cholestasis is a poor prognostic indicator in patients with ALD (Kasztelan-Szczerbinska et al. Alkaline phosphatase: the next independent predictor of the poor 90-day outcome in alcoholic hepatitis. Biomed Res Int. 2013;2013:614081. doi: 10.1155/2013/614081. Epub 2013 Sep 17. PMID: 24151614; PMCID: PMC3789301; Tung BY, Carithers RL Jr. Cholestasis and alcoholic liver disease. Clin Liver Dis. 1999 Aug;3(3):585-601. doi: 10.1016/s1089-3261(05)70086-6. PMID: 11291240; Spahr et al. Early liver biopsy, intraparenchymal cholestasis, and prognosis in patients with alcoholic steatohepatitis. BMC Gastroenterol. 2011 Oct 28;11:115. doi: 10.1186/1471-230X-11-115. PMID: 22035247; PMCID: PMC3228746; Axley et al. Severe Alcoholic Hepatitis: Atypical Presentation with Markedly Elevated Alkaline Phosphatase. J Clin Transl Hepatol. 2017 Dec 28;5(4):414-415. doi: 10.14218/JCTH.2017.00044. Epub 2017 Sep 3. PMID: 29226108; PMCID: PMC5719199)

Comment from Reviewer #1 „What is INR?”

Our response: The abbreviation has been added and explained as follows- the international normalized ratio (INR).

Comment from Reviewer #1 „Ln, 141 it is necessary to equalize the WBC count and RBC count. Abbreviations are missing again.”

Our response: The sentence has been corrected and the abbreviations have been added as follows: red blood cell- RBC count, platelet- PLT count, and white blood cell- WBC count.

Comment from Reviewer #1 „Ln 145 Replace the word conventional.”

Our response: We have replaced the word with routine (markers) instead of using conventional (markers), although the term conventional is widely used by researchers, as we have explained above.

Comment from Reviewer #1 

Comment from Reviewer #1 „I believe that the methods should be classified into several groups. Please form adequate titles and subtitles. You can't write together about blood and serum parameters, then there is a division of patients, certain criteria and symptoms….” 

Our response: We agree with the Reviewer’s assessment. The methods have been corrected and divided into different groups as follows 2.3.1, 2.3.2, 2.3.3, and 2.3.4. Furthermore, the methods have been separated from the patient description (Material).

Comment from Reviewer #1 „Table 2. GTP, please correct this cardinal error. Here it is now ALT, and in methods A1AT. What does the mark kom/uL represent? If the international normalized ratio (INR) is the unit for prothrombin time, there cannot be an INR in the table. It's the same for you if you wrote mmol/L instead of the glucose value. Ln instead of 700 x g specify rpm.”

Our response: We apologize for all the mistakes. They have been corrected as follows: GTP into GGT, kom/uL into cells/uL, and 700 x g to 3000 rpm. 

Comment from Reviewer #1 

Table 4. Please enter a numerical value for each p value, and not combine p and significant levels. Place a mark next to the p value and mark the level of significance (p<0.000) under the table. 

Our response: Our statistical program which is MedCalc® Statistical Software version 20.113 (MedCalc Software Ltd, Ostend, Belgium; https://www.medcalc.org; 2022) for numerical values as low as P < 0.0001 shows only this kind of statistical result. Below we present the result obtained from the program. 

Mann-Whitney test (independent samples)

Sample 1

Variable NE

Filter ALD=1

Sample 2

Variable NE

Filter ALD=0

 Sample 1 Sample 2

Sample size 64 24

Lowest value 69.4154

46.8437

Highest value 637.5770

361.0850

Median 255.1155 107.2220

95% CI for the median 213.1796 to 301.4111 94.3845 to 120.2510

Interquartile range 159.3660 to 380.8010 87.3854 to 138.0075

Hodges-Lehmann median difference -126,8520

95% Confidence interval -187,2400 to -69,5905

Mann-Whitney test (independent samples)

Average rank of first group 52.6406

Average rank of second group 22.7917

Mann-Whitney U 247.00

Large sample test statistic Z 4.881

Two-tailed probability P < 0.0001

Sunday, December 4, 2022 15:42

MedCalc® Statistical Software version 20.113 (MedCalc Software Ltd, Ostend, Belgium; https://www.medcalc.org; 2022)

Comment from Reviewer #1 „Tables 5 and 6 are not described. Remove n from Table 6 to make the table smaller. Please eliminate the green coloring and mark significant differences with *. The description of the images is also not given. It looks like you presented the data as figures next to the table. The figures are not described at all. It is necessary to standardize all tables. It is better to eliminate the green ones, put a sign * and write an explanation for significance under the table. In general, a lot of data in the results is displayed chaotically.

Our response: The Tables and Figures have been corrected according to the Reviewer's suggestions. Table 6-9- n has been removed to make the tables smaller and unify them, the green color has been deleted, statistical significance has been indicated with *, and an explanation for significance has been placed under the tables. 

Also, a description of the figures has been included. 

Comment from Reviewer #1 3.5. There are no data.

Our response: Since no significant differences in the levels of studied neutrophilic biomarkers were observed between patients with or without ALD complications, we just commented on this without showing data. Nevertheless, the data can be presented and they have been included in the revised manuscript text. 

Comment from Reviewer #1 „Discussion

The first page of the discussion (ln 359-385) is a general story. This could have been the Introduction. The discussion is well written.”

Our response: We appreciate the Reviewer’s valuable comments and kind words regarding the discussion. 

We would like to thank Reviewer #1 once again for the comprehensive evaluation of our manuscript. 

Our response to Reviewer #2 (in blue) 

Comment from Reviewer #2 „I have just one recommendation to the authors is :

the type of the study must mention in the abstract”

Our response: We appreciate the time and effort that Reviewer #2 has dedicated to providing valuable feedback on our manuscript. 

The type of our study (prospective one) has been mentioned in the abstract. 

We would like to thank Reviewer #2 once again for the comprehensive evaluation of our manuscript. 

Our response to Editor’s comments (in blue)

Our response: We have checked PLOS One’s style requirements and followed them. 

Our response: The ethics statement has been moved to the Methods section and deleted from the other section.

3. We note you have included a table to which you do not refer in the text of your manuscript. Please ensure that you refer to Table 3 in your text; if accepted, production will need this reference to link the reader to the Table. 

Our response: We have referred to Table 3 in the text of our manuscript.

We would like to thank the Editor for the comprehensive evaluation of our manuscript. 

Kind regards

On behalf of all authors of the manuscript

Beata Kasztelan-Szczerbinska, MD, Ph.D. 

Department of Gastroenterology with Endoscopy Unit

Medical University of Lublin, Poland

---

## [Decision Letter · Decision Letter 1]

20 Dec 2022

Blood concentrations of mediators released from activated neutrophils are related to the severity of alcohol-induced liver damage. -

-

PONE-D-22-28565R1

Dear Dr. Kasztelan-Szczerbińska,

We’re pleased to inform you that your manuscript has been judged scientifically suitable for publication and will be formally accepted for publication once it meets all outstanding technical requirements.

Kind regards,

Gulali Aktas

Academic Editor

PLOS ONE

Additional Editor Comments (optional):

Reviewer comments were addressed adequately in the revised paper. I recommend publication of the manuscript in its current form.

Reviewers' comments:

Reviewer's Responses to Questions

**Comments to the Author**

1. If the authors have adequately addressed your comments raised in a previous round of review and you feel that this manuscript is now acceptable for publication, you may indicate that here to bypass the “Comments to the Author” section, enter your conflict of interest statement in the “Confidential to Editor” section, and submit your "Accept" recommendation.

Reviewer #1: All comments have been addressed

Reviewer #2: All comments have been addressed

2. Is the manuscript technically sound, and do the data support the conclusions?

Reviewer #1: Yes

Reviewer #2: Yes

3. Has the statistical analysis been performed appropriately and rigorously? 

Reviewer #1: Yes

Reviewer #2: Yes

4. Have the authors made all data underlying the findings in their manuscript fully available?

Reviewer #1: Yes

Reviewer #2: Yes

5. Is the manuscript presented in an intelligible fashion and written in standard English?

Reviewer #1: Yes

Reviewer #2: Yes

6. Review Comments to the Author

Reviewer #1: After carefully reading the author's comments, I believe that significant changes have been made in manuscript. I suggest that the manuscript be published.

Reviewer #2: I have just one recommendation to the authors is :

the type of the study must mention in the abstract

7. PLOS authors have the option to publish the peer review history of their article (what does this mean?). If published, this will include your full peer review and any attached files.

Reviewer #1: No

Reviewer #2: No

---

## [Editor Report · Acceptance letter]

27 Dec 2022

PONE-D-22-28565R1 

Blood concentrations of mediators released from activated neutrophils are related to the severity of alcohol-induced liver damage. 

Dear Dr. Kasztelan-Szczerbińska:

I'm pleased to inform you that your manuscript has been deemed suitable for publication in PLOS ONE. Congratulations! Your manuscript is now with our production department. 

Kind regards, 

on behalf of

Professor Gulali Aktas 

Academic Editor

PLOS ONE